# The Importance of Measuring Arsenic in Honey, Water, and PM$_{10}$ for Food Safety as an Environmental Study: Experience from the Mining and Metallurgical Districts of Bor, Serbia

Zorica Sovrlić [1,*], Snežana Tošić [2], Renata Kovačević [1], Violeta Jovanović [3] and Vesna Krstić [1,4,*]

1. Mining and Metallurgy Institute Bor, 19210 Bor, Serbia
2. Faculty of Sciences and Mathematics, University of Nis, Višegradska 33, 18000 Niš, Serbia
3. Faculty of Management Zaječar, Megatrend University Belgrade, 19000 Zaječar, Serbia
4. Technical Faculty Bor, University of Belgrade, 19210 Bor, Serbia
* Correspondence: zorica.sovrlic@irmbor.co.rs (Z.S.); vesna.krstic@irmbor.co.rs (V.K.);
  Tel.: +381-600326550 (Z.S.); +381-668088615 (V.K.)

**Abstract:** Arsenic and toxic metals can reach food and honey through water and air, thus endangering the safe consumption of the same. These toxic substances can damage human health through the food chain, which is contrary to the goals of sustainability related to health and food safety. It is necessary to continuously control and measure the concentration of pollutants to ensure the safety of food produced near mining areas. The arsenic content of honey samples from a territory up to 32 km in diameter from the mine (Bor town, east Serbia, and its surroundings), as determined by ICP-MS, is presented in this paper. PM$_{10}$ particles and water were also analyzed for arsenic content. Pearson's correlation and hierarchical cluster analysis were used for statistical analysis. The obtained results showed that the content of arsenic in honey was below the maximum allowable concentration (MAC) value. The honey was not contaminated, although the results showed that the concentrations of arsenic in water and PM$_{10}$ particles were elevated. The results indicate that the environment in these areas is damaged and point towards caution.

**Keywords:** arsenic; honey; water; PM$_{10}$; food safety; environmental study; pollution

## 1. Introduction

Environmental pollution has been neglected for a long time. Only in recent decades have people realized the adverse effects that industrialization has had on the environment and human health [1,2]. That is how the concept of environmental protection and sustainable development was born. Development that protects the environment is a development that improves social justice [3]. The World Commission on Environment and Development defined this development as "development which meets the needs of the present without compromising the ability of future generations to meet their own needs" [4]. In 2015, the United Nations adopted sustainable development goals, also known as the "global goals". It is a universal call for joint action to eradicate poverty, protect the planet, and ensure peace and prosperity for all people by no later than 2030 [5]. There are a total of 17 goals that are interconnected and which include all aspects of sustainable development—social, economic, and the aspect of environmental protection. For this work, goals number 2 (World without hunger) and number 3 (Good health and well-being) are of special importance. Goal 2 is ending hunger, achieving food security and improved nutrition, and promoting sustainable agriculture. Goal 3 refers to ensuring healthy lives and promoting well-being for people of all generations [5].

Heavy metal contamination of agricultural soils and crops near mining areas is considered a major environmental concern [6]. The results of the research of Stachnik et al. (2020) show data on the distribution of arsenic and heavy metals in Quaternary sediments and

surface waters in a heavily polluted area affected by gold and arsenic mining activity. They show the relationship between sediment and water contamination in a small catchment of the Trująca River. Only arsenic concentration showed a clear increase from the head basin to downstream locations, whereas concentrations of other metals (Zn, Fe, and Al) remained stable [7]. Ng, J.C. et al.'s study assessed the various exposure pathways of arsenic and their health risk apportionment to the residents of Paracatu, a gold mining town in Brazil. Rice and bean were found to contain the highest levels of arsenic, in which the arsenic speciation was measured [8]. The total concentration of As in rice plants (leaves and stems) was linearly correlated to the concentration of As in soil and water. Arsenic contamination of groundwater and soil is a major human health issue, particularly in south and southeast Asia. Use of As-contaminated groundwater from shallow tube wells for the irrigation of paddy rice, the staple food for people in this region, is one of the causes of As-related health impacts [9]. Formenton et al. investigated As pollution originating from the artistic glass industry in Murano. Diarsenic trioxide was a main ingredient of the raw glass mixture until 2015. High arsenic concentrations were recorded in Murano before the sunset date (average 383 ng/m$^3$), representing a serious concern for public health [10]. In Turkey, in the period from 2020 to 2021, water and honey collected in the organized industrial zone in Sivas were tested. Altunay et al.'s research has shown that As present in wastewater from industrial zones is much higher than in wells; the same applies to honey [11].

Many studies have shown that honey can be used as a bioindicator of environmental quality [12–14]. Tong et al. (1975) analyzed honey from an industrial area, a zinc mine, and a major highway in New York State. Some samples from the places near the motorway contained elevated levels of elements such as aluminium, barium, calcium, copper, magnesium, nickel, palladium, and silicon, known to be emitted by traffic [15]. Massidda et al. (2007) used honey as a bioindicator of environmental pollution in an industrial and mining area in Sicily, analyzing the contents of cadmium and lead [16]. Research by Zarić et al. (2022) showed that the highest concentrations of arsenic were determined in honey bees near coal-fired thermal power plants [17]. Maggid et al. (2014) investigated the mercury content of honey produced near mines in Tanzania and the potential risks to public health. The study showed the presence of mercury in the honey samples at the level of contamination, which was below the permissible level. The authors concluded that the honey sampled from selected villages in Tanzania was suitable for human consumption. The authors pointed to the necessity of future research in the vicinity of the mine to determine the sites of contamination, identify the extent of contamination of and exposure to mercury in other foods, and recommend measures to minimize exposure to mercury in Singida and other areas in Tanzania [18].

Anthropogenic influence on the environment results in a decrease in honey bee populations worldwide, with varying degrees of morbidity and mortality [19]. Today, bees face numerous problems, primarily due to industrialization and human activity [20–22]. The decline of the global bee population threatens the natural pollination process, which can have disastrous consequences for both the planet and humans. The contribution of bee pollination in promoting sustainable development goals through food security and biodiversity is widely recognized [23]. Honey, bee pollen, propolis, royal jelly, beeswax, and bee venom are used in traditional and modern medicine. The mineral composition of honey comes from plants, which receive the minerals from soil and water. Solayman et al. (2016) confirmed that the mineral composition of honey also comes from environmental pollution, so it is necessary to preserve the environment and prevent its contamination [24]. Bee colonies are especially threatened near mining facilities due to the accumulation of heavy metals [25].

Eastern Serbia is known for its ore deposits. The ore deposits are distributed in the wider region, within the Carpathian–Balkan zone, i.e., in the Timok eruptive massif of eastern Serbia. One of the largest mining and metallurgical complexes in this area is the mining and smelting basin in Bor, now Zijin. Đurđevac-Ignjatović et al. (2022) discussed the issue of environmental protection as a consequence of pollution and waste generation from

the Bor smelter and its contribution to the achievement of sustainable development goals in that part of Serbia [26]. From the beginning, copper production in Bor has been followed by the environmental pollution of soil, water, air, and vegetation, with air pollution being recognized as the most dangerous [27]. Matić et al., analyzing the data of the Serbian Environmental Protection Agency, the Mining and Metallurgical Institute of Bor, and the Institute of Public Health in Belgrade, Serbia, indicated the presence of As and Cd in ambient air $PM_{10}$ near industrially contaminated locations. That is because ores have a high content of heavy metals and metalloids, a public health problem in the Republic of Serbia. According to the research of these authors, As concentrations in $PM_{10}$ are above the limit value in Bor and Lazarevac, while Cd values are slightly increased in Bor [28].

Bor is one of the most polluted cities in Europe, which affects the population's health. The impact of heavy metals transported by air to the environment and ending up in the food chain has not been investigated. However, the rise in the contents of As, Cu, and Pb has been registered in some plants. Additionally, increased concentrations of As and Pb have been recorded in some workers of the Zijin Bor company and the residents of Bor, Serbia [29,30].

This paper shows the results of As concentration in honey as well as As in water and air in the zone of the copper mine in the city of Bor in eastern Serbia. Mining exploitation and copper processing in Bor have seriously damaged the environment. Twenty-three locations in the Bor region were included in the honey testing. Honey was sampled in the diameter of 4 to 32 km from the source of copper mine pollution. In order to better understand the obtained results of the present concentration of As in honey, the presence of As in rivers near the 23 locations from where the honey was taken was sampled and analyzed. In addition, the content of As was also analyzed in $PM_{10}$ particles, which were placed and sampled in analyzers in city locations closest to the mine.

## 2. Experimental Part and Methodology

### 2.1. Experimental Part

Honey samples were collected from May to September 2016 in the territory of Bor, a district of eastern Serbia, and were prepared by microwave digestion according to the USEPA 3052 method [31]. The arsenic content was analyzed using ICP-MS with the Agilent 7900. Microwave dissolution of the samples was performed using an ETHOS 1 microwave system (Milestone, Bergamo, Italy) containing an SK 12 rotor. Dissolution was performed under the following conditions: power of 1000 W, pressure 40 bar, temperature of 180 °C for 15 min; 10 mL of 65% $HNO_3$ (Merck, Darmstadt, Germany) and 2 mL of 10% $H_2O_2$ (Merck, Darmstadt, Germany) were used to dissolve the sample. After cooling, all samples were filled to a volume of 50 mL and measured using ICP-MS. The sampling sites are marked with numbers from 1 to 23, and the types of honey are marked with the letters A and M, representing acacia and meadow honey, respectively. The names of the mentioned locations and types of honey are shown in Table 1.

In addition to determining the arsenic content in honey samples, the electrical conductivity of a 20% honey solution and the pH of a 10% honey solution (manufactured by Inolab) were measured according to the methods recommended by the International Honey Commission [32]. A digital conductometer Eutech instrument, PC 10, was used to determine electrical conductivity, and a WtW 154 pH meter was used for pH measurements.

Quartz filters, Whatman QMA, 47 mm in diameter, were used for sampling respirable $PM_{10}$ particles. Two samples of zero filters were used to ensure quality during the measuring process. A filter was randomly selected from each box and measured for 5 consecutive days (laboratory zero filter). A field zero filter is used to assess possible contamination during the process of putting on, removing, and transporting the filter. The field zero filter is a filter that, like all others, is placed in its holder and taken to the field, placed in the insert of the sampler while the pump is not turned on, kept for 15 min, removed from the insert, returned to its holder, and then returned to the laboratory, where it is subject to a conditioning procedure, similar to all the other filters. Unlike the field zero filter, the

laboratory zero filter, after measuring the empty filters, remains in the laboratory to assess the conditions during the measurement [33].

The AMS Park automatic measuring station is part of the network for the systematic monitoring of air quality in Bor, with the aim of collecting data on basic and specific pollutants in order to research the impact of air quality on human health, climate, and forest ecosystems in order to take appropriate preventive measures.

The dissolution of deposits on the exposed filter was performed by the microwave digestion method under the standard CEN/TC 264 N 799 (2006) [34]. For this, Merck chemicals of high ultra-pure grade and double-distilled water (MilliQ, 18.2 M$\Omega$) were used.

Using a microwave oven (the Milestone model ETHOS™ EASY, Sorisole, Italy, with 12 cuvettes), the destruction of samples of respirable particles was performed in a two-stage temperature program at 200 °C. Under the same conditions, a sample of the certified reference material NIST 1648a was devastated by microwaves according to the procedure of the chemical analysis of respirable particles.

### 2.2. Methodology and Main Characteristics of Honey

Acacia honey has a distinctly light yellow color, with a mild, pleasant smell and taste. It remains in a liquid state for months. It is one of the types of honey that crystallize very slowly because it contains more fructose than glucose. Figure 1 was created using the QGIS version 3.22.10 program, Gary Sherman, Odense, Denmark.

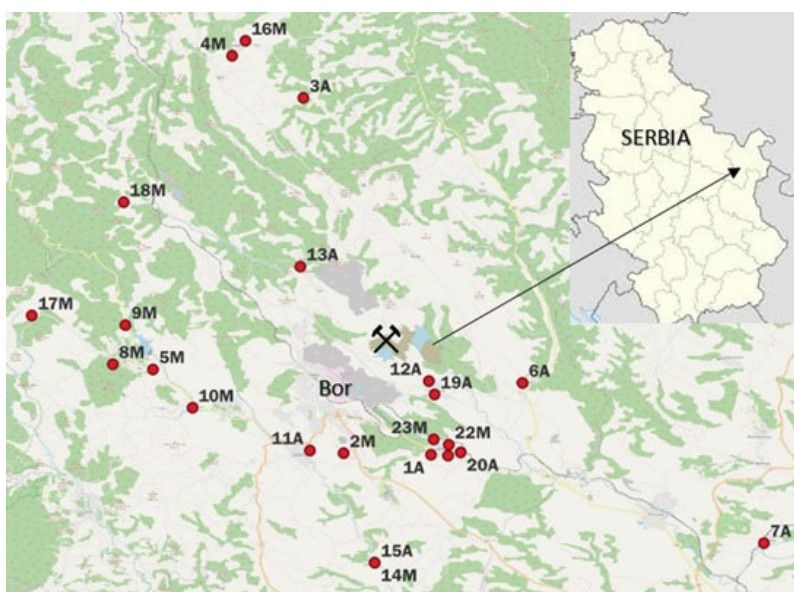

**Figure 1.** Sampling sites. The locations where the hives are located are indicated by ordinal numbers around the mine ZiJin, Bor.

Meadow honey is honey made from various meadow flowers, so it has a wide range of effects. Judging by the amount of meadow honey produced in Serbia, meadows are important pastures for bees. For this reason, it is necessary to keep meadows unpolluted as much as possible in the vicinity of mine exploitation. Additionally, it is important to analyze acacia and meadow honey from the territory of the mining activity of copper exploitation and processing and monitor the possibility of the contamination of honey with As. To get the content of As in honey, we determine the presence of As in water and air (PM$_{10}$ particles). Pearson's correlation study and hierarchical cluster analysis were performed using the IBM SPSS 20 statistical package (IBM, Armonk, NY, USA).

## 3. Results

### 3.1. As in Honey and Water

The inorganic part of honey consists of the number of elements from the periodic system of elements. The mineral composition of the studied honey shows the presence of macro elements such as Na, K, Ca, Mg, P, S, and Cl and trace elements such as Al, Cu, Pb, Zn, Mn, Cd, Tl, Co, Ni, Rb, Ba, Be, Bi, U, V, Fe, Pt, Pd, Te, Hf, Mo, Sn, Sb, La, I, Sm, Tb, Dy, Sd, Th, Pr, Nd, Tm, Yb, Lu, Gd, Ho, Er, Ce, Cr, As, B, Br, Cd, Hg, Se, and Sr. The results are of the same order of magnitude as published by Solayman et al. (2016) [24]. As is present in honey in traces, but when it appears in an elevated amount, it can cause serious health problems.

The results of As content in honey are given in Table 1. It has already been mentioned that honey monitoring was chosen according to the beehives' locations so that monitoring would cover the surrounding zone of the copper mine in Bor, which is considered one of the most polluted zones in eastern Serbia. Despite the pollution of the zone, the results show that the concentrations of As do not exceed the permissible limits in any sample of honey.

**Table 1.** The concentration of As in bee honey. pH and electrical conductivity of honey.

| Location | | Distance from the Source of Pollution (km) | As in Honey mg/kg | pH of Honey | Elec. Conductivity of Honey µS/cm |
|---|---|---|---|---|---|
| Symbol | Name | | | | |
| 1A | Slatina | 5.44 | 0.0110 | 3.52 | 163.7 |
| 2M | Novi Gradski | 2.28 | 0.0540 | 3.89 | 467.0 |
| 3A | Tanda | 15.9 | 0.0037 | 3.28 | 378.0 |
| 4M | Gornjane | 18.71 | 0.0157 | 3.82 | 658.0 |
| 5M | Savača | 7.25 | 0.0662 | 3.76 | 892.0 |
| 6A | D.B. Reka | 7.67 | 0.0132 | 3.38 | 246.0 |
| 7A | Trnavac | 19.53 | 0.0033 | 3.57 | 145.3 |
| 8M | TilvaNjagra | 8.95 | 0.1219 | 3.98 | 1017.0 |
| 9M | Borsko Jezero | 8.83 | 0.0622 | 4.02 | 954.0 |
| 10M | Brestovačka Banja | 5.79 | 0.0702 | 3.76 | 676.0 |
| 11A | Brestovac | 4.05 | 0.1137 | 3.68 | 358.0 |
| 12A | Oštrelj | 3.85 | 0.0148 | 3.55 | 129.0 |
| 13A | Krivelj | 6.21 | 0.0359 | 3.62 | 191.5 |
| 14M | Džanovo Polje | 10.08 | 0.1919 | 3.93 | 599.0 |
| 15A | Džanovo Polje | 10.08 | 0.0672 | 3.45 | 166.9 |
| 16M | Gornjane | 19.07 | 0.0151 | 3.98 | 492.0 |
| 17M | Crni Vrh | 12.62 | 0.1505 | 4.11 | 776.0 |
| 18M | Mali Krivelj | 13.28 | 0.0316 | 4.07 | 684.0 |
| 19A | Oštrelj | 4.24 | 0.0034 | 3.63 | 143.0 |
| 20A | Slatina | 6.67 | 0.0265 | 3.52 | 141.6 |
| 21M | Slatina | 6.28 | 0.0491 | 3.79 | 428.0 |
| 22M | Slatina | 6.13 | 0.1763 | 4.09 | 612.0 |
| 23M | Slatina | 5.29 | 0.1167 | 3.99 | 584.0 |

Adamović et al. (2021) [35] published detailed results on the elevated presence of As, $SO_4^{2-}$, Fe, Cu, and Mn in river waters located in the Bor mining area in Eastern Serbia, in the same zone where As was analyzed in honey and $PM_{10}$ particles. The total concentrations of As in unpolluted river water, according to Adamović et al. (2021) [35],

were up to 5 μg/L. The authors showed that As concentrations in polluted river waters collected from the Krivelj and Bela rivers were elevated compared with As concentrations in river water samples collected in unpolluted areas. For example, the total As concentrations in samples along the Bela Reka, according to Adamović et al. (2021) [35], were 395, 573, 715, 373, and 528 μg/L. In the vicinity of this river, there were beehives marked with sample 6A, where the concentration of As in honey was 0.0132 mg/kg, which can be seen in Table 1. The authors also published total As concentrations along the Kriveljska River with the values of 1.5, 15, 98.7, and 57.3 μg/L. Near this river, there were beehives from which honey was sampled that had the sample mark 13A. In this case, the concentration of arsenic in the honey was 0.0359 mg/kg, Table 1. In general, multi-year measurements of arsenic concentration in rivers indicate the existence of an increased concentration of arsenic in the water in the mine area.

The permitted amount of 0.5 ppm, given by the Rulebook on the quantities of pesticides, metals, and metalloids and other toxic substances that can be found in foodstuffs; it can also be observed that the As content is slightly higher in meadow honey than in acacia. These values for meadow honey are expected, and they differ due to the different types of plants and the nectar that bees can collect; therefore, the content of active substances that can be found in this type of honey is higher [36].

To determine the present correlations, Pearson's correlation analysis was performed. The results show a good correlation (Pearson's correlation coefficient 0.608) between the content of As in honey and the pH value of honey. The correlation between As in honey and the distance from the mine is weakly negative (−0.220), which is expected, namely, that with a greater distance from the mine, there will be less arsenic in the honey. A positive correlation between As in honey and electrical conductivity exists (0.520); a positive correlation also exists between electrical conductivity and pH (0.747).

Živkov-Baloš (2021) pointed out that the composition of honey includes more than 20 organic acids (tartaric, citric, oxalic, acetic, etc.), which amount to about 0.50% of the honey [37]. The presence of acids and the pH value of the honey depend on the types of plants from which the honey is produced and on the geographical location. Acacia honey had a lower pH value that ranged from 3.28 to 3.68, which represents the minimum values, in contrast to the meadow honey samples, whose pH values ranged from 3.76 to 4.11 units, representing the maximum values, from the 23 locations where the honey samples were taken.

In order to obtain reliable data on plant origin, honey is analyzed according to its physical and chemical properties. One of the physical and chemical properties is the specific electrical conductivity, which can also be used to identify honey. Honey conducts electricity better when the content of present mineral salts is higher. The obtained results of honey samples show that there is a difference in electrical conductivity values (Table 1) between acacia and meadow honey. Acacia honey had a lower electrical conductivity, ranging from 129 to 378 μS/cm, unlike the meadow honey samples, whose electrical conductivity reached as high as 1017 μS/cm.

The hierarchical cluster analysis of the samples based on the contents of As shown in Figure 2.

Two main clusters can be observed on the dendrogram. One contains the majority of meadow honey samples, and the other contains two subclusters, one containing the majority of the meadow honey samples and the other containing the majority of the acacia honey samples. In general, most of the acacia honey samples are in one cluster, while the meadow honey samples are distributed into two groups that do not belong to the same cluster; this leads to the conclusion that meadow honey is more relevant for biomonitoring.

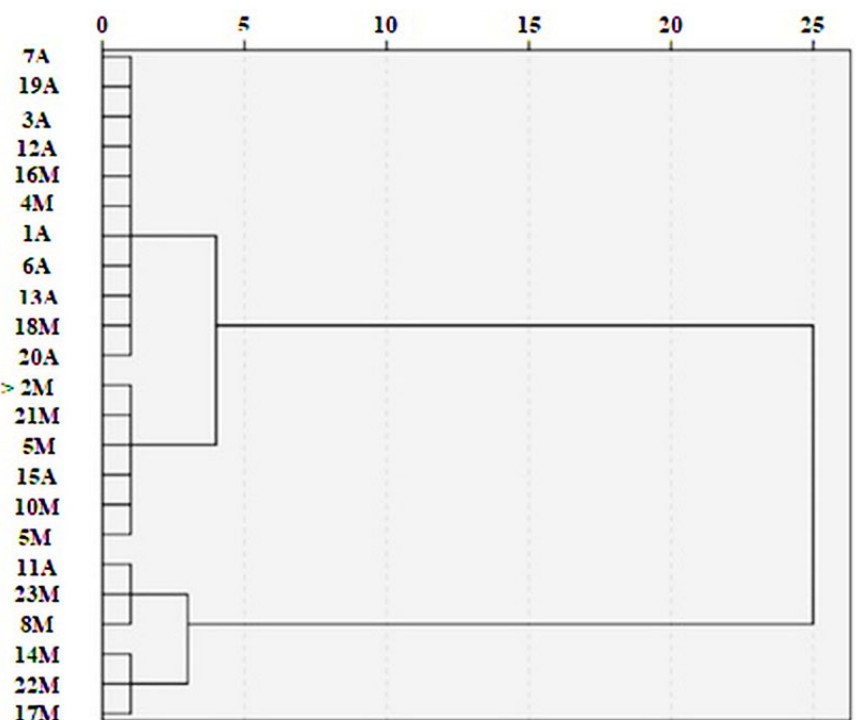

**Figure 2.** Hierarchical dendrogram for the investigated samples based on As content.

### 3.2. As and Accompanying Metals in $PM_{10}$ Particles

The primary objective of air monitoring is to determine the extent to which the contents of various pollutants in the air are in accordance with the values prescribed by the corresponding air quality standards. In different countries of the world, the regulations have different values. According to the current EU regulation, Council Directive 2008/50/EC [38], the average daily value for $PM_{10}$ is 50 $\mu g/m^3$. The prescribed value must not be exceeded for more than 35 days a year. The allowed mean value at the annual level for $PM_{10}$ is 40 $\mu g/m^3$ [39,40].

Figure 3 shows the monitoring of seasonal air pollution using $PM_{10}$ particles, the content of As, and the accompanying metals (Pb, Cd, and Ni). Usually, the contribution from the burning of wood and coal is characteristic of the winter period, and most of the fugitive dust and fires are tied to the summer period. Industrial emissions do not have pronounced seasonal fluctuations but are the result, first of all, of the operating regime of the industrial plant and metrological parameters.

Based on the data of $PM_{10}$ particles in Figure 3, a Pearson analysis was also performed; a good correlation between $PM_{10}$ and As (0.629) as well as Cd and As exists (0.730).

As can be seen from Figure 3a, the median value for $PM_{10}$ during the measuring period (2016) is 27.4 $\mu g/m^3$. This value is lower than the allowed mean value at the annual level of 40 $\mu g/m^3$ [38,39]. Minimum and maximum values for $PM_{10}$ were 0.9 and 161.2 $\mu g/m^3$, respectively. For Pb, the median value was 59 $ng/m^3$, which is lower than the allowed mean value of 500 $ng/m^3$, with minimum and maximum values of 3.0 and 637.0 $ng/m^3$, respectively.

The median value for Ni is 4.0 $ng/m^3$, which is lower than the allowed mean value at the annual level of 20 $ng/m^3$, with minimum and maximum values of 0.3 and 80.3 $ng/m^3$, respectively (Figure 3b). Median concentrations of As and Cd were higher than the maximum allowable values. For Cd, the median value was 1.18 $ng/m^3$ (5 $ng/m^3$ allowed mean value at the annual level), and for As, the median value was 34.1 $ng/m^3$ (6 $ng/m^3$ allowed mean value at the annual level). These data indicate elevated concentrations of As in $PM_{10}$ at the measuring site, which was also shown by the Pearson's correlation coefficient; this probably indicates the same source of pollution. Šerbula et al. (2021) and Tasić et al. (2017)

indicated in their studies that almost all the obtained results show increased concentrations of As in PM 10 [41,42].

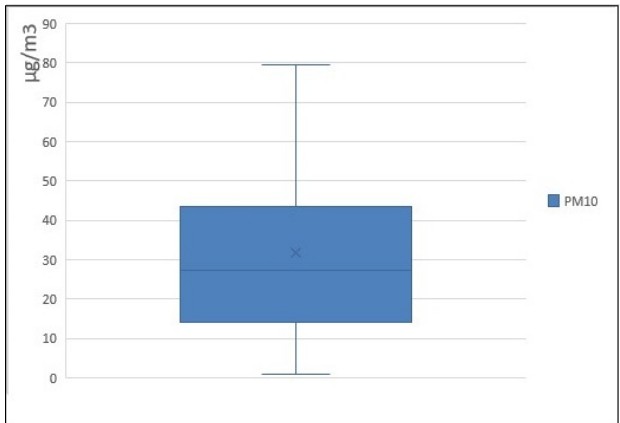

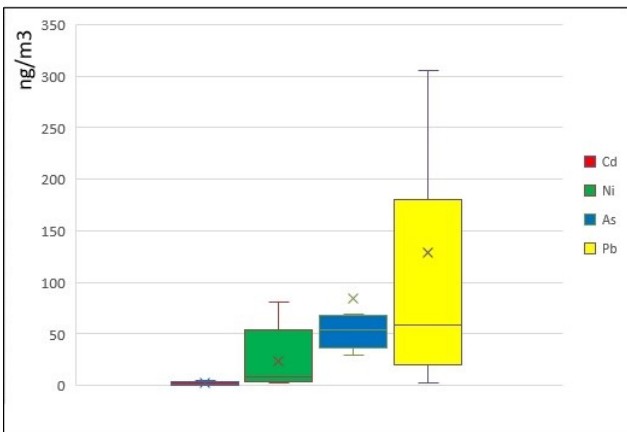

**(a)**

**(b)**

**Figure 3.** Boxplot (**a**) the content of $PM_{10}$ particles ($\mu g/m^3$) and boxplot (**b**) the content of As, Cd, Pb and Ni ($ng/m^3$) at the measuring site Gradski Park during 2016.

Yadav et al. and Kulshrestha et al. showed that the contribution from motor vehicles, secondary nitrates, and fugitive dust was noticeably higher during weekdays compared to weekends. On the other hand, a higher contribution of industrial emissions, liquid fuel combustion, and secondary sulfates was recorded during weekends [43,44]. Observing the difference between weekdays and weekends, a greater contribution of motor vehicles and resuspended dust is observed during weekdays, which clearly indicates their origin from traffic. In contrast, the contribution from burning wood and coal shows the opposite trend, with higher concentrations during weekends (Figure 4). The phenomenon can be explained by the fact that during weekends, people spend more time indoors in their apartments and consume a larger amount of wood for heating during the winter season, as confirmed by the results in Figure 4.

Arsenic is a known marker for the non-ferrous metal industry, as shown by Yli-Tuomi et al. (2003) [45]. Gidhagen et al. (2002) identified significant amounts of arsenic, up to 30.7 $ng/m^3$ in $PM_{10}$ particles, at seven rural locations between 10 to 100 km from the nearest copper and gold smelters [46].

The composition of ore and the ore processing process determine the primary types of emissions from the copper smelter. To illustrate, emissions from several copper smelters in Chile are characterized by different types of pollution. Hedberg et al. (2005) analyzed $PM_{10}$ particles collected near a copper mine in Chile; the presence of As, Cu, and Zn ions was confirmed from the emissions of the Caletones smelter. The presence of As, Cu, Mo, and Ag was also confirmed from the emissions of the Ventana smelter. The presence of As, Cu, Mo, and Bi in $PM_{10}$ particles was confirmed from the emissions of the Chagres smelter [47].

Table 2 shows the results of the presence of As in the matrices related to bees and honey. The results of different authors show a high level of As concentration in different test areas. Comparing the results of As in honey in the Bor district and honey products in other parts of the world can help us to see the real danger to human health when consuming honey. High-temperature processes of the roasting and melting of ores in smelters for the production of copper in Bor in eastern Serbia, smelting operations in foundries, waste burning, various operations in the cement industry, etc., emit easily volatile pollutants such as As, Cd, Pb, and Ni, as confirmed by the results shown in Figure 4. Additional fuel combustion during the smelting process also contributes to the emission of respirable $PM_{10}$ particles.

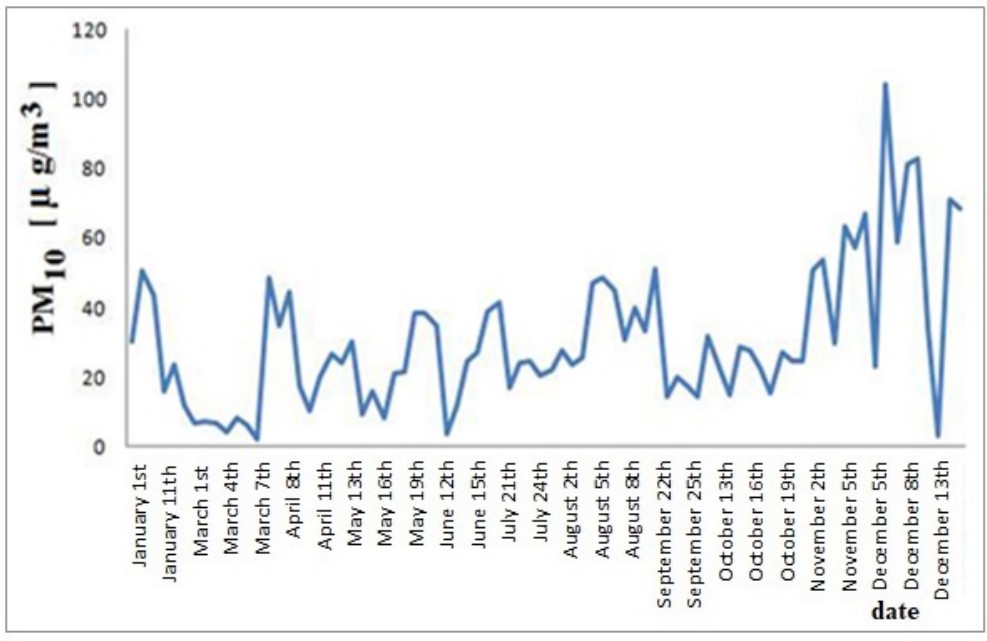

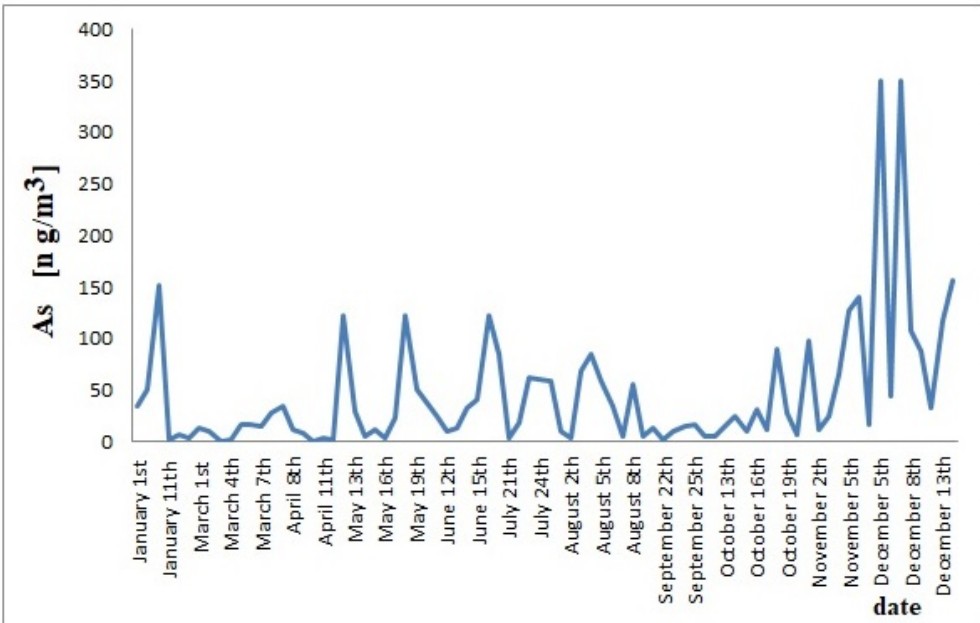

**Figure 4.** Time monitoring of $PM_{10}$ and the presence of As at the measuring site Gradski Park for 2016.

**Table 2.** As concentrations in the different matrices in previous studies.

| Matrix | As ($\mu$g g$^{-1}$) | Comments | Ref. |
|---|---|---|---|
| honeybee matrix | 0.00456 | Injection volume: 20 L (5 L of palladium chloride and magnesium nitrate were added as a regulatory mixture) | [48] |
| | 0.00151 | | |
| | 0.00446 | | |
| | 0.00284 | | |
| adult honeybees | <0.50–12.50 | 72 sites (rural–urban) | [49] |
| | <0.10 | Hives without CCA [1] | [50] |
| | 0.67–0.83 | to large industrial areas | [51] |
| propolis matrix | 146.24 | Argon, 250; 30 s; 110 °C; 1 °C s$^{-1}$ | [48] |
| | 43.55 | Argon, 250; 30 s; 130 °C; 15 °C s$^{-1}$ | |
| | 578.46 | Argon, 250; 20 s; 1250 °C; 10 °C s$^{-1}$ | |
| | 28.81 | Argon, 250; 3 s; 2250 °C; 1 °C s$^{-1}$ | |
| | 18.99 | Argon, 250; 3 s; 2450 °C; 1 °C s$^{-1}$ | |
| adult honeybees | 0.77–1.11 | Hives with CCA 3 urban site close | [50] |
| honey samples from Cankırı City, Turkey | 14.7 ± 4.7 | Mean ± standard deviation (n = 7, k = 4) | [52] |
| honey samples ($\mu$g kg$^{-1}$) | 13.83 ± 1.90 | [2] LOD = 0.54 $\mu$g L$^{-1}$ [3] LOQ = 1.8 $\mu$g L$^{-1}$ [4] BEC = 0.35 Spiked sample s= 3.232 $\mu$g g$^{-1}$ | [53] |
| | 4.23 ± 0.06 | | |
| | 1.70 ± 0.01 | | |
| | 1.70 ± 0.01 | | |
| | 1.70 ± 0.01 | | |
| Pollen samples ($\mu$g kg$^{-1}$) | 361.30 ± 18.88 | [2] LOD = 0.0036 $\mu$g L$^{-1}$ [3] LOQ = 0.012 $\mu$g L$^{-1}$ [4] BEC = 0.13 Spiked samples = 3.163 $\mu$g g$^{-1}$ | [53] |
| | 330.16 ± 4.49 | | |
| | 93.21 ± 25.16 | | |
| | 51.10 ± 22.19 | | |
| | 7.00 ± 0.45 | | |
| meadow honey from Bor, Serbia, with the highest As conc. ($\mu$g kg$^{-1}$) | 191.9 | pH = 3.93; el. cond. = 599.0 $\mu$S cm$^{-1}$ | This work |
| | 150.1 | pH = 4.11; el. cond. = 776.0 $\mu$S cm$^{-1}$ | |
| | 176.3 | pH = 4.09; el. cond. = 612.0 $\mu$S cm$^{-1}$ | |
| | 116.7 | pH = 3.99; el. cond. = 584.0 $\mu$S cm$^{-1}$ | |

[1] CCA—chromated copper arsenate (wood preservative substance); [2] LOD—limit of detection; [3] LOQ—limit of quantification; [4] BEC—background equivalent concentration.

## 4. Discussion

The amount of harmful substances that irreversibly escapes from chimneys into the atmosphere depends on many factors. Some of them are the choice of technological procedures for copper ore processing, the composition of the input raw material, the temperature and duration of the process, and the type and amount of process gases. All technological procedures used in copper ore processing emit As, and other accompanying elements (Cu, Pb, Cd, Ni, Zn, Bi, Sb) are distinguished and stand out from other toxic substances.

According to the data of Ilic et al. [54], in the period from 1991 to 2001, Mining Complex RTB Bor emitted between 5.3 and 19.6 kg of As per inhabitant of the municipality of Bor on an annual basis. Bearing in mind that the content of arsenic is extremely high and that the EPA has classified inorganic arsenic as a Group A human carcinogen [55], special attention should be paid to understanding the processes in which this element is formed and separated, considering that it is the main source of the presence of As in water, soil, and air, and from there, honey.

For this reason, there is a multi-decade monitoring of environmental pollution with heavy metals. The SEPA Agency (Environmental Protection Agency, Ministry of Envi-

ronmental Protection, Republic of Serbia) [56] monitors soil, water, and air pollution on a daily, monthly, and annual basis, where the data has been visible online since 2006. Antonijević et al. (2012) found that the soil is very polluted with copper, iron, and arsenic in this area [57].

The content of arsenic in copper ores from the Bor ore deposits is usually low. The average content of arsenic in the copper concentrate processed in the smelter in Bor is between 0.05% and 0.15%. A slightly higher content of arsenic is contained in the boron concentrate, 0.5–0.8%. Before the smelting process itself, several types of concentrates and solvents (quartz and limestone) are usually mixed so that the final arsenic content in the obtained batch does not exceed 0.5%.

During pyrometallurgical smelting, arsenic and its compounds are converted from flame furnaces into gases that, after purification by a dry process, are released through chimneys into the atmosphere, polluting the environment of the town of Bor and its surroundings.

Laboratory tests on the melting of the boron concentrate with a high content of arsenic (2.73–3.29%) have shown that the degree of arsenic evaporation increases with increasing temperature and extensions in frying time. Subsequent industrial tests have shown that with an increase in the batch frying temperature (arsenic content between 0.13% and 1.56%) to a temperature of 100 K, the volatility of arsenic increases by about 9%. Each sample of honey is an extremely complex mixture of very different chemical ingredients. Impeccably pure honey that is produced and processed differently can show great differences in chemical properties. The properties, composition, and changes in the content of certain basic chemical substances of honey can only be understood by comprehensive observation of the life of bees and the quality of the environment [20,58].

Matin et al. (2016) analyzed bees and their products (honey, propolis, wax) in an industrial zone in Turkey as good biological indicators of environmental pollution [48]. The reasons that bees, and therefore bee products, are used for the biomonitoring of environmental pollution are because they are easy to breed and are widely distributed in nature and their body is covered with hair. This allows honeybees to retain and transport different materials they come into contact with all the way to the hives; they have a high reproduction rate, high mobility, and a wide flight range that allows them to cover a large area of flight for feeding [59].

Bastias et al. [12] aimed to determine in their study the concentration of total and inorganic As in honey samples collected over a three-year time period in different areas of Chile. The Valenar area in the Atacama region, which is characterized by a high level of mining activity, was also included in the research. The results showed that the highest concentration of arsenic was found in honey samples from these areas. After longer analyses, Bastias et al. finally concluded that the total concentrations of As do not pose a risk to human health because the levels of As found are below the permissible limits [12]. In their study, Bratu and Georgescu (2005) used honey produced in apiaries in Copşa Mica (Sibiu region, Romania), known for the ecological imbalances caused by the non-ferrous metal industry. Amounts of heavy metals (Pb, Cd, and Zn) above the permissible limit contained in honey samples collected in the mentioned area were determined [60]. Boryło et al. (2019) examined the radioactivity of honey from northern Poland. The results of the research indicated an increased content of polonium in certain areas that are associated with the development of many branches of industry (e.g., the chemical industry, the petrochemical and sodium industries, as well as the fertilizer industry). Finally, the authors concluded that honey consumption is safe and has no negative impact on human health [61]. Pipoyan et al. [62] indicated that Cu concentrations were above the maximum allowed level in several types of honey from Syunik, a region in Armenia owned by the mining industry. Additionally, the values of nickel and arsenic exceeded the permitted level of carcinogenic risk values, and the author's conclusion was that this may be a cause for concern [62]. To determine the potential risks posed by old abandoned mines, Álvarez-Ayuso and Abad-Valle (2017) compared soil, honey, and pollen samples from those areas with uncontaminated areas. The results showed that the soil around the mine had slightly

elevated levels of Cd and Sb (about 2 to 3 times the normal soil concentration), while the concentration of As reached significant levels, with concentrations almost ten times higher than what is considered toxic [63].

Toxic metals and metalloids present in the environment, including As, can be deposited together with pollen on the hairs on the body of bees and thus reach the hives; they can also be adsorbed together with nectar from flowers or water. Several factors can significantly affect the monitoring of heavy metals that affect bees and their products, such as weather conditions (rainfall and wind rose), seasons, botanical origin of the honey, etc. Porrini et al. (2003) noted that acacia honey is cleaner because its flower faces downwards and is less exposed to precipitation and air pollution, which is not the case with meadow honey, where there is diversity in terms of plants and their nectar and the modes of air and water pollution [64].

Pollution by toxic elements is a known environmental problem in regions where mining, industry, and agriculture are developing. These areas are characterized by heavy metals in soil, water, and air. Plants can absorb soil pollutants and heavy metals and enter the nutrient cycle. Heavy metal contamination raises environmental concerns, such as entering the food chain and contaminating food, which is harmful to human health. Consumption of food contaminated with heavy metals can cause various health disorders. Bees also absorb heavy metals by consuming contaminated water, pollen, and nectar, such as inhaling particles during the summer period when they are most active. Environmental pollution can negatively affect the safe consumption of honey and possibly lead to risks to human health, which is contrary to the goals of sustainable development. Therefore, the monitoring of bee products in terms of toxic elements in regions with a pronounced mining industry is very important for food safety and for the prevention of potential future environmental problems in order to achieve the goals of sustainable development [65].

In general, the results of previous studies have mainly indicated an increased concentration of heavy metals and arsenic in honey produced near mining areas; however, in most cases, they did not exceed illegal limits [12], and hence, honey is safe for use and human health [66,67]. However, some authors have called for caution [18,63,68,69], which is also the case with the authors of this study.

## 5. Conclusions and Future Perspectives

This paper presents the presence of As in samples of acacia and meadow honey, surrounding waters, and ambient air (PM$_{10}$ particles) in the area of copper mines in eastern Serbia. Based on the obtained results, it is evident that there is no excess content of As in any analyzed sample of honey. It means that there is no elevated values of As in honey near the source of pollution nor at a greater distance from the mine. The results show that the control of As in bee honey can serve as an indicator of honey quality; however, it is not a reliable bioindicator of environmental pollution in eastern Serbia, where the increased presence of As in water and PM$_{10}$ particles has been registered.

It is known that honey is the least contaminated bee product, which was confirmed by this research. Future research could focus on the analysis of other bee products, such as propolis, wax, and pollen, with the aim of using these honey products as potential bioindicators of environmental contamination. Helpful information would be obtained about their usefulness in human nutrition and medicinal properties in human use, depending on the pollution of the territory where the honey is collected. Such research is of great importance for the study of the environment because it enables the safe consumption of food and the preservation of human health. Additionally, the research will indicate the quality of the environment and provide the necessary data for making important decisions to preserve the environment to achieve sustainable development.

**Author Contributions:** Conceptualization, Z.S., V.J. and V.K.; methodology, Z.S., S.T., V.J. and V.K.; validation, S.T.; formal analysis, S.T.; investigation, Z.S.; resources, Z.S. and R.K.; data curation, S.T.; writing—original draft preparation, V.K.; writing—review and editing, V.K., and Z.S.; visualization, R.K., V.J. and S.T.; supervision, V.K. and S.T.; project administration, Z.S. and S.T.; funding acquisition, Z.S. All authors have read and agreed to the published version of the manuscript.

**Funding:** Financial support for this study was provided by the Ministry of Science, Technological Development and Innovation of the Republic of Serbia (contract No. 451-03-47/2023-01/200052).

**Institutional Review Board Statement:** Not applicable.

**Informed Consent Statement:** Not applicable.

**Data Availability Statement:** The data presented in this study are available on request from the corresponding author.

**Conflicts of Interest:** The authors declare no conflict of interest.

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
