# Peer review of "The Importance of Measuring Arsenic in Honey, Water, and PM10 for Food Safety as an Environmental Study: Experience from the Mining and Metallurgical Districts of Bor, Serbia"

_sustainability, doi:10.3390/su141912446_

Round 1

Reviewer 1 Report

Dear Authors,

This research is done very good. The results are useful and valuable for reporting.

I have some comments for revision of this manuscript:

1. The manuscript has some typographical errors. Please read the manuscript carefully and correct it completely.

2. Please add some new references (2020-2022) and compare your results with recently published articles especially if you can find similar researches in Serbia. 

3. The quality of figures is low, please insert high quality figures especially Fig 1, 2 and 5. 

4. Conclusion and perspective is too long. Please revise this part. In  conclusion, it is better you mention about the main results. Also for perspective, there are some unnecessary statements. Try to shorten this part.

Author Response

Answer to Reviewer 1

Thank you very much for your useful comments.

This research is done very good. The results are useful and valuable for reporting. I have some comments for revision of this manuscript:

  1. The manuscript has some typographical errors. Please read the manuscript carefully and correct it completely.

Each sentence has been carefully read and grammar and spelling have been corrected.

  1. Please add some new references (2020-2022) and compare your results with recently published articles especially if you can find similar researches in Serbia. 

Added 8 references from recent years. Added references are visible in the Reference List and commented in the text, which is also visible

  1. The quality of the figures is low, please insert high-quality figures, especially Fig 1, 2 and 5. 

Fig. 1 was removed from the text at the request of Reviewer 2, and Fig. 2 and Fig. 5 have been corrected and inserted into the text of the Manuscript.

  1. The conclusion and perspective are too long. Please revise this part. In  conclusion, it is better you mention the main results. Also for perspective, there are some unnecessary statements. Try to shorten this part.

The conclusion has been carefully revised according to your instructions so that it is now non-obstructive.

Best Regard,

Zorica Sovrlić & Vesna Krstić, et al.

Reviewer 2 Report

The topic of the article is interesting and original. The article should be accepted for publication in the selected journal. but it is necessary to check the text in advance - mainly misspellings and misspellings, as well as spaces between words (e.g. in the authors' names - Vesna Krstič, etc.).

Example of typo: Abstract - no politants but pollutants, etc.Please check the full text.

PM10 - particle size should be expressed with a subscript. Please check the full text.

What is MDK in the abstract? Please explain in words and then use the abbreviation.

Introduction: the article is mainly about arsenic, but lines 47-67 are about other elements. Try to add relevant studies if they exist.

In my opinion, Figure 1 is unnecessary.

In the acknowledgments section, you have the editor's instructions, not your own text. Please rewrite or cross out, etc.

Author Response

Answer to Reviewer 2

Thank you very much for your nice comments.

  1. The topic of the article is interesting and original. The article should be accepted for publication in the selected journal. but it is necessary to check the text in advance - mainly misspellings and misspellings, as well as spaces between words (e.g. in the authors' names - Vesna Krstič, etc.).

Technical errors have been corrected and indicated in the comments on the right side of the Manuscript.

  1. Example of typo: Abstract - no politants but pollutants, etc.Please check the full text.

Thanks for the comment. Each sentence has been carefully read and grammar and spelling have been corrected.

  1. PM10 - particle size should be expressed with a subscript. Please check the full text.

Carefully done.

  1. What is MDK in the abstract? Please explain in words and then use the abbreviation.

Done. Visible in the Abstract.

  1. Introduction: the article is mainly about arsenic, but lines 47-67 are about other elements. Try to add relevant studies if they exist.

The entire paragraph has been replaced with new references. Visible in the Introduction.

  1. In my opinion, Figure 1 is unnecessary.

Figure 1 was removed.

  1. In the acknowledgments section, you have the editor's instructions, not your own text. Please rewrite or cross out, etc.

Sorry, the Acknowledgment was deleted.

Best Regard,

Zorica Sovrlić & Vesna Krstić, et al.
